# Ethical care in patients with Covid-19: A grounded theory

Hamideh Azimi[1], Rafat Rezapour-Nasrabad[2]*, Fariba Borhani[3], Akram Sadat Sadat Hoseini[4], Fariba Bolourchifard[3]

1 Student Research Committee, School of Nursing and Midwifery, Shahid Beheshti University of Medical Sciences, Tehran, Iran, 2 Department of Psychiatric Nursing and Management, School of Nursing and Midwifery, Shahid Beheshti University of Medical Sciences, Tehran, Iran, 3 Department of Medical Surgical Nursing, School of Nursing and Midwifery, Shahid Beheshti University of Medical Sciences, Tehran, Iran, 4 Pediatrics and NICU Department, School of Nursing and Midwifery, Tehran University of Medical Sciences, Tehran, Iran

☯ These authors contributed equally to this work.
* rezapour.r@sbmu.ac.ir

**Data Availability Statement:** All relevant data are within the manuscript and its Supporting Information files.

**Funding:** We are heartily grateful to the research council of the Shahid Beheshti University of

## Abstract

### Background

Providing ethical care during the Covid-19 pandemic has become an inevitable challenge due to facing limitations such as fear of contracting the disease, lack of equipment and emergence of ethical conflicts; So that there is no clear picture of how to provide ethical care for patients with Covid-19. The study aimed to explain the ethical care process of patients with Covid-19.

### Method

This qualitative study was conducted in 2021–2023 using the grounded theory research method. Data were collected through conducting 21 semi-structured interviews with 19 participants (16 staff nurses, and 3 supervisor). Sampling was started purposively and continued theoretically. Data analysis was performed by the method proposed by Strauss and Corbin.

### Results

The results indicated that starting the process with a problem means a challenge of how to do the right or correct thing for the patient. This process is driven by the feeling of duty and inner commitment to do the right thing. The sense of responsibility or commitment is directly related to the degree of faith in a person. The actions that a nurse takes to fulfill her sense of commitment is called the faith-based behavior process. The process of behavior based on faith is done through two steps. Attaining the satisfaction of God and as well as the satisfaction of one's conscience is the nurses' ultimate goal of doing the right work and being committed.

Medical Sciences for financially supporting the study.

## Conclusion

The process of ethical care of patients with Covid-19 can be different according to the degree of faith and environmental conditions. A higher level of faith and more helpful environmental conditions, the greater the nurse's inner commitments in the first and second steps, as a result, the satisfaction of God and conscience will be greater.

## Introduction

Ethics, according to intellectuals, means any personality trait that causes good or bad behaviors [1]. Ethics can also refer to the issues, experiences, views, or behavioral systems of a group. For example, Christian ethics means the behavioral system accepted by Christians [1]. Accordingly, the science of ethics can be considered a science that defines a framework by expressing the value and necessity of acquired traits and optional actions in order to achieve a desirable life [1, 2]. With this description, ethics can be considered the basic pillar of the nursing profession [3], because nursing care and interventions are a subset of optional tasks [4]. Therefore, commitment to ethics in the matter of care is the central core of nursing values and forms the basis of nursing practice and the framework of nurses' behavior with patients [3]. Non-observance of ethical values in care not only has a significant, negative impact on patients [5], such as delaying recovery, increasing the possibility of physical and mental injuries [6], the non-observance of patient rights, and ultimately reducing patient satisfaction [7], it will also have negative consequences for nurses, including the triggering of anger, demoralization, decreased motivation, and an unwillingness to stay in the nursing profession [8].

Compliance with ethical principles and values is necessary at all times, but in some situations such as the Covid-19 pandemic, maintaining compliance is very difficult because of the special conditions of the patients. Nurses face limitations in caring for Covid-19 patients, such as fear of contracting the disease or transmitting it to other family members [9], the decision-making of the treatment staff instead of the patient [10], physicians' lack of accurate information about treatment methods, and the recovery process and care ambiguities following it [11]. Treatment staff may encounter torment of conscience following the uncertainty of choices, the compulsion to perform actions that conflict with ethical values and personal beliefs, damage to the principle of human relations [10], confusion, helplessness, and ambiguity as a result of the lack of an ethical functional framework in the care of these patients [12, 13], unwillingness to care for patients, and distancing from patients because of the possibility of infection [10] and the stigma caused by it [14]. All of these situations can lead to many ethical conflicts regarding the care of these patients, some of which have not ever been observed in similar diseases so far [14]. By undermining bioethical principles such as justice, autonomy, beneficence, and non-harm, the provision of ethical care during the Covid-19 pandemic has become an inevitable challenge [10].

The results of a 2020 study by Ferorelli indicated the necessity of providing an ethical framework in which to make any decision about care for patients with Covid-19 [15]. In their 2020 study, Fink showed that quarantining patients, not allowing family visits, and not allowing a companion to be with Covid-19 patients, not even for disabled patients, has created ethical conflicts among medical staff regarding how to care for such patients [16]. Huxtable conducted a study in 2020 in England and stated that the lack of a national ethical framework for providing proper care to patients with Covid-19 is a care vacuum and a cause of confusion for nurses [17]. Despite the advancement of ethics in the field of healthcare, little is known about ethical care, how to provide it, what solutions and behavioral patterns to use in facing

ethically challenging situations or obstacles, and facilitators of providing ethical care [18]. Therefore, more studies are needed to better understand how to provide ethical care to patients, identify the process of ethical care, and deeply understand the conditions and contextual factors affecting ethical care.

Undoubtedly, nursing knowledge can be enhanced by an in-depth identification of operational processes for providing ethical care. In other words, knowing the right path, having the right ethical reasoning, and finally choosing the correct action according to the situation the nurse has experienced are effective factors in minimizing ethical dilemmas in the care of these patients [3]. To determine how to provide ethical care to Covid-19 patients and identify the factors affecting it, experiences should be examined and understood from the perspective of the participants. Moreover, there is a need to identify how nurses can interpret and choose solutions for different situations. Using grounded theory with deep and rich data generation [19] is the best way to realistically explain the process of ethical care for Covid-19 patients. Therefore, considering the processual nature of ethical care, the lack of clarity about the dimensions and manner of ethical care in accordance with the ethical problems of Covid-19, and the lack of investigations into the process of providing ethical care to this group of patients in the world, the present study aims to explain the process of ethical care of Covid-19 patients.

## Methods

### Study design

This qualitative study was conducted in 2021–2023 using the grounded theory research method and the analytical approach of Strauss and Corbin (2008).

### Participants and data collection

The study population comprised nurses with clinical work experience of 6 months or more who agreed to share their experiences with us. The study was conducted in hospitals treating Covid-19 inpatients in the cities of Qazvin and Tehran, Iran. Purposeful sampling was begun by the first author and included nurses in different treatment, educational, and managerial positions who worked or had worked in Covid-19 inpatient wards, were willing to provide information, and spoke Farsi. After data analysis, the emergence of concepts and primary categories, and the recording of notes, the primary categories were completed by selecting the next participants by theoretical sampling with maximum diversity. During theoretical sampling, one should attend places, persons, and situations that provide the maximum opportunity to discover more and better new concepts, identify relationships between them, and determine how concepts change under different conditions [19]. In other words, based on the concepts resulting from the data, one should seek to obtain more information from the concepts and complete the categories that are being formed [19]. It is not possible to determine the number of participants before the study is conducted. Rather, the theoretical saturation of the categories and the evolving theory is considered as an indicator for determining sample size and completing sampling [19]. Theoretical saturation occurs when no new perspective or category and sub-category are created from the coding process [19]. Theoretical sampling in the present study continued until theoretical saturation of the categories and the emergence of the core concept and connecting it with other categories were achieved. In the present study, after 19 interviews, theoretical saturation of categories occurred. This means that no new category was obtained in the last two interviews; but to ensure theoretical saturation, two more interviews were also conducted. Thus, 21 interviews with 19 participants was enough to complete the categories and determine the grounded theory in the present study. It should be noted that in

order to complete the categories and clarify some ambiguities in the interviews, two participants were interviewed twice.

The main method of data collection was semi-structured in-depth interviews. Before conducting each interview, the interviewer attempted to gain the trust of the interviewee by introducing themself, stating the study objectives, and explaining how the interview would be conducted. The first interview was conducted as a pilot and after its analytical review by the research team, more appropriate questions for the next interview guide were developed. The first version of the interview guide is available within the manuscript (S1 Table). The interviews started with a general and open-ended question: "Please describe one day of your experience in caring for a patient with Covid-19." Based on the interview guide, the answers provided by the participants, and the researcher's recorded notes, more specific questions were then asked. After analyzing each interview, the interview guide was revised to conduct more in-depth interviews. To properly guide the flow of the interview and complete the information, the researcher used techniques such as probing by leading, asking the participant to tell more, echo probing, posing long questions, uh probing, graceful interruption, nodding and smiling based on the process of the participants' answers. Field notes such as pauses, laughter, eye contact, etc. were also attached to the text of the interviews. Because of quarantining and the emergency conditions caused by the Covid-19 pandemic in Iran, the interviews were conducted in a combination of face-to-face and web-based forms. The time and place of the interviews were determined according to the wishes of the participants, and with their written consent, all interviews were recorded and typed verbatim. Interviews with participants were conducted from 16 September 2021 to 15 November 2022.

In addition to using the interview method, the unstructured observation method was also used to obtain more field notes, match the items stated in the interviews, and record the non-verbal behaviors of the participants. The researcher used face-to-face rather than web-based interviews, made appointments in the corona ward, and spent several shifts as a nurse's assistant in the corona ward to obtain unstructured observations of nurses' behavior with patients. Important observations during the interview and the researcher's presence in the corona ward were noted in the form of headings and points, and immediately after the interview and leaving the hospital, the notes were written and completed in more detail. Data collection and analysis were done simultaneously.

## Data analysis

Data were analyzed concurrently with data collection using the approach proposed by Corbin and Strauss (2008). This approach consists of the four steps of data analysis for concepts, data analysis for context, data analysis for process, and integration of categories. In the first step, the first author carefully transcribed the recorded interviews and then reviewed the written interviews and notes several times to achieve a proper understanding of the text. Then, the first author broke the data into manageable parts, after which pieces of data were explore in terms of the ideas that were inside them. The first author then gave these ideas a conceptual label (coding). A list of primary codes was created, and the primary codes were compared with each other for similarities. Codes with conceptual similarity were grouped into primary categories. Throughout the coding and analysis process, the researcher recorded the research team's ideas about the data, themes, and emerging conceptual framework in the form of notes or mental dialogue between the researcher and the data.

After categorizing the different codes and placing them in primary categories, these categories were also re-categorized according to their characteristics and dimensions as well as the conceptual relationship between them, and were included in more abstract categories. In this

step of the analysis, the researcher used some of the techniques mentioned by Corbin and Strauss (2008), such as constant comparison, theoretical comparisons, questioning of data, the flip-flop technique, and use of personal experiences to increase theoretical sensitivity.

While analyzing the data for concepts and creating conceptual and abstract categories, the second step was also accomplished (i.e., analyzing the data for context in terms of micro and macro conditions and the process of people's response to the conditions). In this step, the researcher examined situations and conditions that affected people's actions or feelings and caused their responses to differ. By asking questions such as why, who, what, where, when, and under what social and environmental conditions around the concept of ethical care, the researcher analyzed the root of people's actions and tendencies.

In the third step (data analysis for the process), the researcher used the questioning technique to determine the main concept and the main process in the data and how the participants managed their main problem. By answering questions like "What is going on here?" or "What's going on?" or "How do you care for patients with Covid-19?" the researcher started writing the descriptive storyline of each interview around the main concept.

In the last step (i.e., the integration of categories), the researcher reviewed the storyline related to each interview several times and created a conceptual map that included all the narratives. Finally, by organizing the conceptual map, the narrative of the story was completed. MAXQDA18 software was used to organize and manage data.

### Rigor and trustworthiness

Trustworthiness was ensured using the 10 criteria recommended by Strauss and Corbin [19]. How each criterion is achieved is explained in Table 1.

### Ethical consideration

This study was approved by the ethics committee of Shahid Beheshti University of Medical Sciences (IR.SBMU.PHARMACY.REC.1400.106). After that the written agreement of the administrators of the study setting was got before starting the study. Electronic or written informed consent to participate in the study and audio recording was obtained from all participant. Minimal and relevant data set as the underlying data for this study are available within the manuscript (S1 File).

## Results

The demographic characteristics of the participants are presented in Table 2.

The 19 participants comprised 15 women and 4 men.

### "Faithful nursing" theory

The data-based theory known as the "faithful nursing theory" provides an outline of the process that nurses go through after encountering a patient with Covid-19. With their unique personality traits, nurses provide care for Covid-19 patients under all the environmental threats and the support relevant to the emergency situation of the Covid-19 pandemic. This process begins with nurses' main concern, i.e., the challenge of how to do what is necessary and correct for their patients. This process is driven by the feeling of duty and commitment to do the right thing. Nurses describe their task-orientation as the factor motivating them to take action. As much as nurses consider themselves obligated and feel commitment, they strive to do the right thing; therefore, the fulfillment of commitment is graded. The sense of responsibility or

**Table 1. Rigor and trustworthiness.**

| criterion | How to achieve criteria | |
|---|---|---|
| Fit | Fit with experiences through: Reviewing concepts obtained by some participants as well as two nurses who did not participate in the study (Member check); Reviewing interviews, categories, and concepts obtained by three professors with experience in the field and then checking their opinions about the extent to which the findings are believed. | |
| Applicability or Usefulness | Offering new explanations or insights in ethical care through: Selecting diverse participants, clearly and accurately describing the characteristics of participants (age, education, work experience), providing a detailed description of the results using quotations. | |
| Concepts | Developing common understandings through: Findings would be organized around concepts, studying related texts and articles, deepening the data and meanings of concepts, having multiple meetings with supervisors and advisors (peer review), obtaining the opinions of four experienced professors in qualitative research (external review). | |
| Contextualization | Explaining the findings in the context of personal and environmental conditions through: Giving quotes for the reader to fully understand the events that happened, describing the events along with the existing environmental conditions in detail. | |
| Logic | Logical connection between concepts and not having a feeling of emptiness and ambiguity in the course of the study through: Clearly presenting the method and providing details of data collection and analysis, drawing shapes and diagrams of the logical flow of ideas. | |
| Depth | Deepening the findings through: Conducting in-depth interviews, clarifying ambiguous and multidimensional cases in interviews by re-interviewing, providing a detailed description of the findings during data analysis and interpretation, having multiple meetings with supervisors and consultants to review all the codes, concepts, and notes to extract hidden dimensions in findings such as the existence of faith. | |
| Variation | Showing complexity and diversity through: Combining data collection methods (interview, observation, notes), interviewing participants who had maximum diversity to record different experiences, paying attention to patterns that seem to conflict with the general pattern, such as the effect of instincts negative on the way of providing care, placing conflicting patterns as the basis of theoretical sampling and proposing other questions to further examine the issue and develop concepts in terms of dimensions and characteristics. | |
| Creativity | Having a new understanding of the findings through: Having a new insight and worldview in data analysis, getting guidance from experienced supervisors and consultants in discovering new concepts such as jihadi management, being the first study in the field to explain the ethical care process of patients with Covid-19. | |
| Sensitivity | Being sensitive to reactions and statements through: Continuous comparison, theoretical comparison, flip-flop technique, thinking in multiple meanings of a word, using personal experiences as a nurse's assistant in the Corona ward, participating in medical ethics workshops, reviewing various studies in the field of philosophy of medical and nursing ethics and grounded theory, identification of indicators of nursing and medical ethics in various studies, extraction of questions necessary to collect data after analyzing each interview. | |
| Evidence of memos | Using memos through: Using all perceptions and thoughts recorded by the researcher from the beginning of the analysis through the entire analysis process, placing them for theoretical sampling, and proposing appropriate questions in the interview guide to deepen the interviews. | |

**Table 2. The characteristics of the participants and the interviews.**

| | |
|---|---|
| Length of interviews | 45–76 min |
| Ward | intensive care units, emergency, medical, neurological, infectious |
| Work experience in hospital | 1–24 years |
| Work experience in Covid-19 | 6–31 months |
| Age | 23–46 years |
| Participants | Staff nurse: 16; supervisor: 3 |
| Educational degree | PhD degree: 1; master's degree: 2; bachelor's degree: 16 |

commitment is directly related to the degree of faith in a person; the stronger the faith of nurses is, the more robust is their feeling of commitment. Therefore, the core concept of the faithful nursing theory is a process called "behavior based on faith", which represents the actions that nurses take to manage their main problem and resolve their concern.

The process of behavior based on faith is implemented in two steps (Fig 1) depending on the context. It is affected by the existing environmental conditions and the personality characteristics of each nurse. A higher level of faith and more helpful environmental conditions will not only make proceeding through these two steps easier, but will also reduce the problems caused by a patient's non-observance of ethical values while simultaneously improving the nurse's sense of satisfaction. The grounded theory process of "faithful nursing" is explained in more detail in the next section.

### The challenge of how to do the right thing for the patient as an inner commitment

The main concern of nurses at any moment is to do what they think is right and necessary for the patient; such concern arises for nurses from the manifestation of their sense of duty and commitment. Commitment is the call of the human conscience and reason, which is always consistent with human nature. Nevertheless, this commitment is an internal matter and not an external requirement. Nurses considering it their duty to perform a necessary task for a patient, one for which there is no external law or requirement, in the face of environmental threats and difficulties, is an indication of the existence of an inner requirement or force which causes nurses to voluntarily consider themselves committed. The 16th participant said about the lack of external obligation: *"There were things that were the doctor's duty, but if there was a*

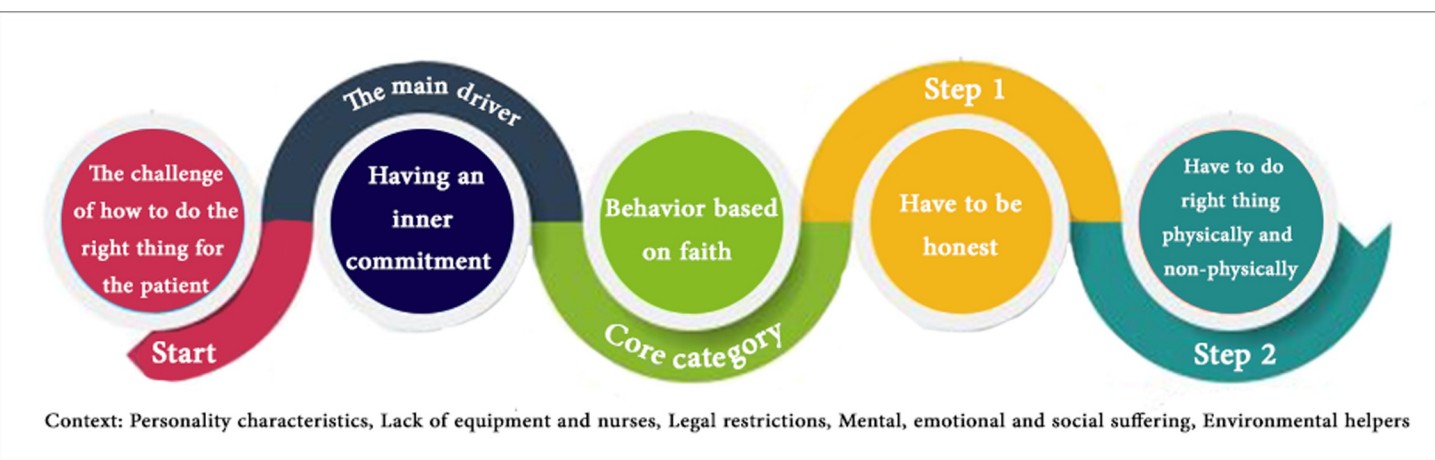

**Fig 1. Faithful nursing: A grounded theory of behavior process based on faith.**

*delay, the patient might get hurt, so I accepted this responsibility so that the patient wouldn't get hurt" (P16)*. The seventh participant referred to fulfilling her commitment under the mental difficulties and sufferings related to the actions of the human mind, such as thinking, remembering, etc., saying: *"We were bothered a lot. There were many times when I came home and I was constantly thinking about them, and it bothered us a lot, but still I didn't give up on my work" (P7)*. Another participant complained of the existence of social sufferings related to the ridicule and stigmatization of colleagues for doing the right thing: *"I have been told for a while that I am a nosy lady. I tickle the patient so much that something comes out of him. They not only do not encourage you, but worse, they destroy you" (P1)*.

Nurses sometimes experienced cases where they had not only no legal obligation, but also a legal prohibition to do what they thought was right, but they performed the task anyway, because they thought it was right. The 13[th] participant said about disobeying the law in order to do what she considered necessary for the patient: *"There was a patient who had edema and his lungs were throbbing. We informed the doctor, but we knew that it would take a long time for the doctor to come. We made a decision to give him Lasix, even though we didn't have a doctor's order" (P13)*. In addition to the existence of laws that prevent them from doing the right thing, the existence of contradictory laws caused confusion for nurses. This legal challenge not only caused a serious disruption in the care process, but was also one of the differences between the care of Covid-19 patients and other patients.

In addition to the aforementioned difficulties, restrictions such as a lack of equipment and too few nurses also impeded the work of nurses. One of the participants said: *"If my patient called me before, I would go to him 80% of the time, but now maybe 50% of the time. Well, we have a shortage of nurses and they don't provide nurses" (P3)*. The low number of nurses has always been considered a major limitation and obstacle to providing proper patient care; the hindrance of this restriction was doubled during the Covid-19 pandemic. On the one hand, nurses became infected with the Covid-19 disease and had to go on leave; on the other hand, some nurses requested to be transferred from corona inpatient wards or quit work due to physical or family problems. *"At the beginning, we had colleagues whose wives were fighting with them. Some of our colleagues resigned. How many lives have been lost because of this corona" (P1)*. Clearly, doing what one thinks is right and feels committed to do, despite such difficulties, indicates the presence of a strong and inner motivation in a person.

Fortunately, nurses also had helpers who facilitated the provision of care. For example, one participant said: *"To the nurse who had a busy patient this shift, I would give a low-needs patient in the next shift, give her an incentive, or when the ward was quieter, I would say she didn't need to come" (P14)*. Another factor of taking care of nurses was the creativity and innovation of managers in implementing strategies to compensate for the lack of nurses and equipment. Some methods were novel and unique and had never been mentioned in any treatment situation other than Covid-19 in Iran. One method was the use of jihadist groups and non-medical volunteer forces. *"In the hospital, there were a large number of sick patients who were not accompanied, and all the primary work of these patients was with the nurses. After some time, jihadist groups came. With their arrival, the pressure was removed from the nurses. They came to feed patients and to talk to them. They took some pressure off the staff" (P16)*. Another initiative is related to the hourly admission of patients to keep beds empty, which is stated for the first time only in this study. Another way to solve the problem of nursing and equipment shortages was to use contract nurses or request nurses and equipment from other departments and hospitals. Despite the hardships and suffering endured by the nurses and helpers, these people consider themselves committed to doing what they consider necessary and right for the patient as far as possible.

### Faith is the manifestation of commitment

Manifestation of such a sense of commitment in nurses originates from the faith that exists in their hearts. Faith in Arabic is Iman. Faith is not just a word but a heart-felt belief, one that is a force directing human life and playing an important role in how one lives. It is the focus of valuing a person's thoughts and actions. Faith is the inclination of the heart and intellectual, religious, and spiritual dependence on a superior being, a powerful creator, and a saving philosophy.

Faith can be the result of a combination of a person's values, experiences, and desires. In every inner conflict and challenge, one of these three elements prevails and influences the manifestation of the feeling of commitment. For example, one participant explained the positive effect of her beliefs on her behavior, saying: *"Once in a while you see something unsterilized, so that belief makes you go in a good direction, but it may be very troublesome. For example, changing a set of serum or infusion sets with burettes takes a lot of time" (P13)*. Participant fifth said about the impact of her experimental observations: *"I saw that patients have less respiratory distress when they are relaxed; the same relaxation would increase their saturation by 5–6 degrees. That's why I tried to calm my patients" (P5)*. On the other hand, every nurse can have positive and negative desires and instincts, which can naturally and uniquely affect their choice of behaviors. Participant 13th described her stress and the destructive effect it has on caregiving: *"Maybe my stress can have an impact. For example, if my patient's condition worsens, I will quickly lose myself" (P13)*.

One interesting point is that this inner feeling of commitment and the faith behind it has not only natural roots, but also an extraterrestrial and God-like aspect. In other words, considering God's satisfaction was evident in the nurses' behavior. By attributing their commitment to the satisfaction of God, not only were the sanctity and importance of faith increased in nurses, but the final destination becomes clearer in their behavioral framework. In this regard, the 12th participant said about doing her inner commitment and relating it to God: *"I believe in God and my conscience is very important to me. That means I don't like to do less work at all. The issues of the rights of the people and being accountable to God are very important to me" (P12)*. Therefore, the existence of the conscientious origin of commitment as well as its divine attribute, the nurses' ultimate goal of doing right and being committed can be considered attaining the satisfaction of God as well as the satisfaction of one's conscience.

### Behavior based on faith

When nurses are faced with the challenge of how to do the right thing for a patient with Covid-19, based on the degree of faith they have, they consider themselves committed to doing what is necessary and right for the patient. This sense of commitment is fulfilled through actions called behavior based on faith. As shown in Fig 2, behavior based on faith progresses through two steps, the second having five sub-paths (guide, jihad, heal grief, encourage to continue living, create peace of mind) to reach the final destination. Each sub-path is described in more detail in the next section and in Table 3.

**Being honest.** As shown in Fig 3, the first step of behavior based on faith is feeling committed to being honest in both speech and behavior. Nurses tried to answer their patients' questions in various ways while being truthful and expressing that truth in an appropriate way. Therefore, it must be taken into account that Covid-19 differed from other diseases in that its treatment process, effective drugs, and patient prognoses were unknown. Although these factors caused nurses to be troubled by telling the truth, through their initiative, they were always honest. For example, the 13th participant said: *"I used to talk in two ways. For example, I said that it is impossible to say that the corona drugs are either completely effective or completely*

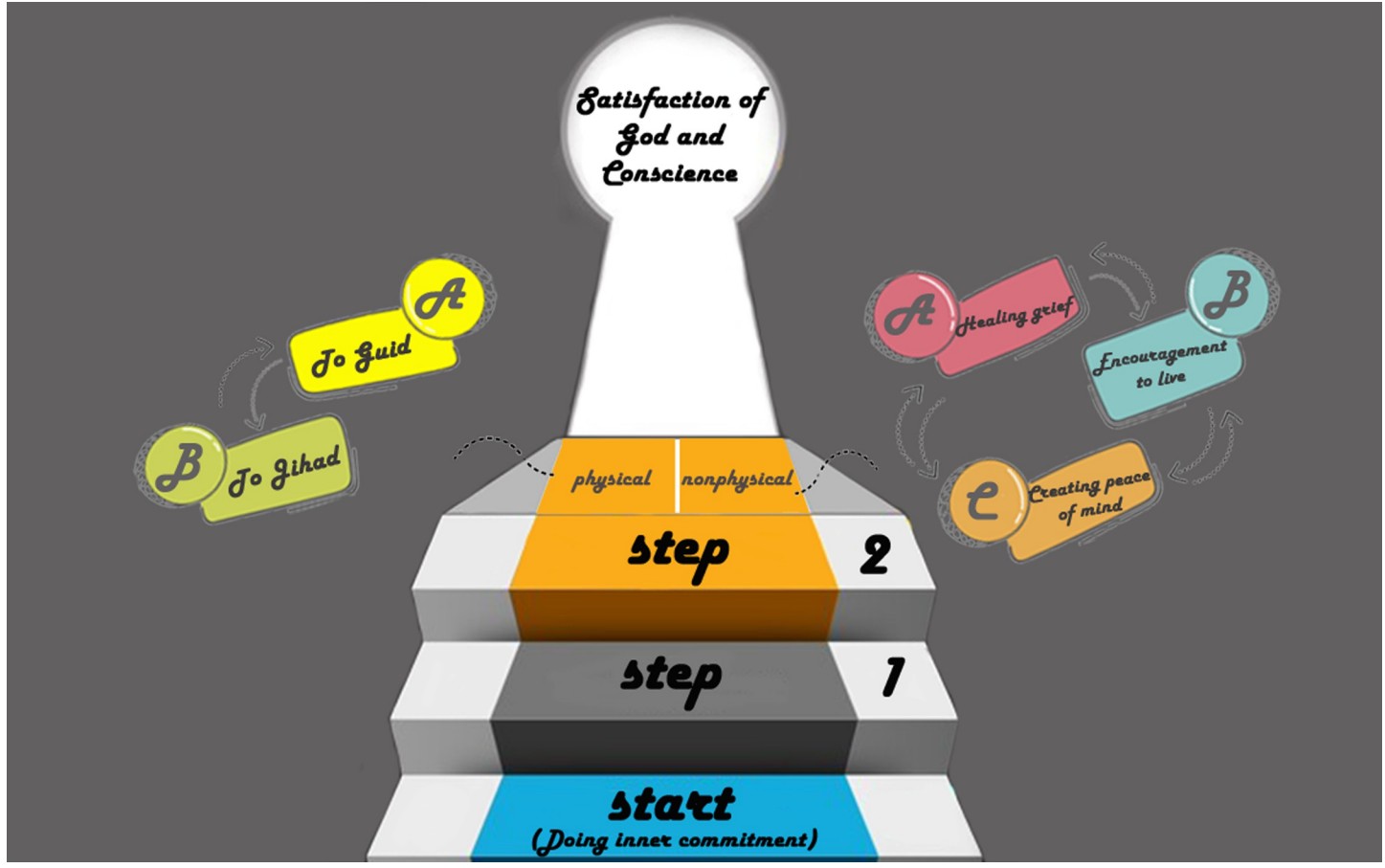

**Fig 2. The process of behavior based on faith.**

*ineffective. Instead, a series of conditions work together. That is, I say it in two ways until the patient himself understands" (P13).*

Honesty in the behavior of nurses is also a fulfillment of their. In other words, the adherence of nurses to their professional obligation was the biggest sign of their honesty in behavior, because nurses are concerned with providing care efficiently and in accordance with the correct principles in order to be accountable in their work and present no harm to the patient.

**Table 3. Five sub-paths in the second step in the process of behavior based on faith.**

| Guide | Jihad | Heal grief | Encourage to continue living | Create peace of mind |
|---|---|---|---|---|
| Patient and family awareness | Restraint in difficulties | Emotional compatibility | Improving the vitality and energy of the patient | Respect for patient privacy |
| Convince the patient | Patience and tolerance | Gentleness | Creating life expectancy in the patient | Respect for the patient's beliefs and opinions |
|  | Sacrifice | Talking with the patient | Spiritual help to the patient | Efforts to gain the patient's trust |
|  | Observance of justice | Creating physical comfort for the patient |  | Eliminate the patient's fear and anxiety |
|  | Moral courage | Helping the companions of the patient |  |  |
|  | Financial assistance to the patient |  |  |  |

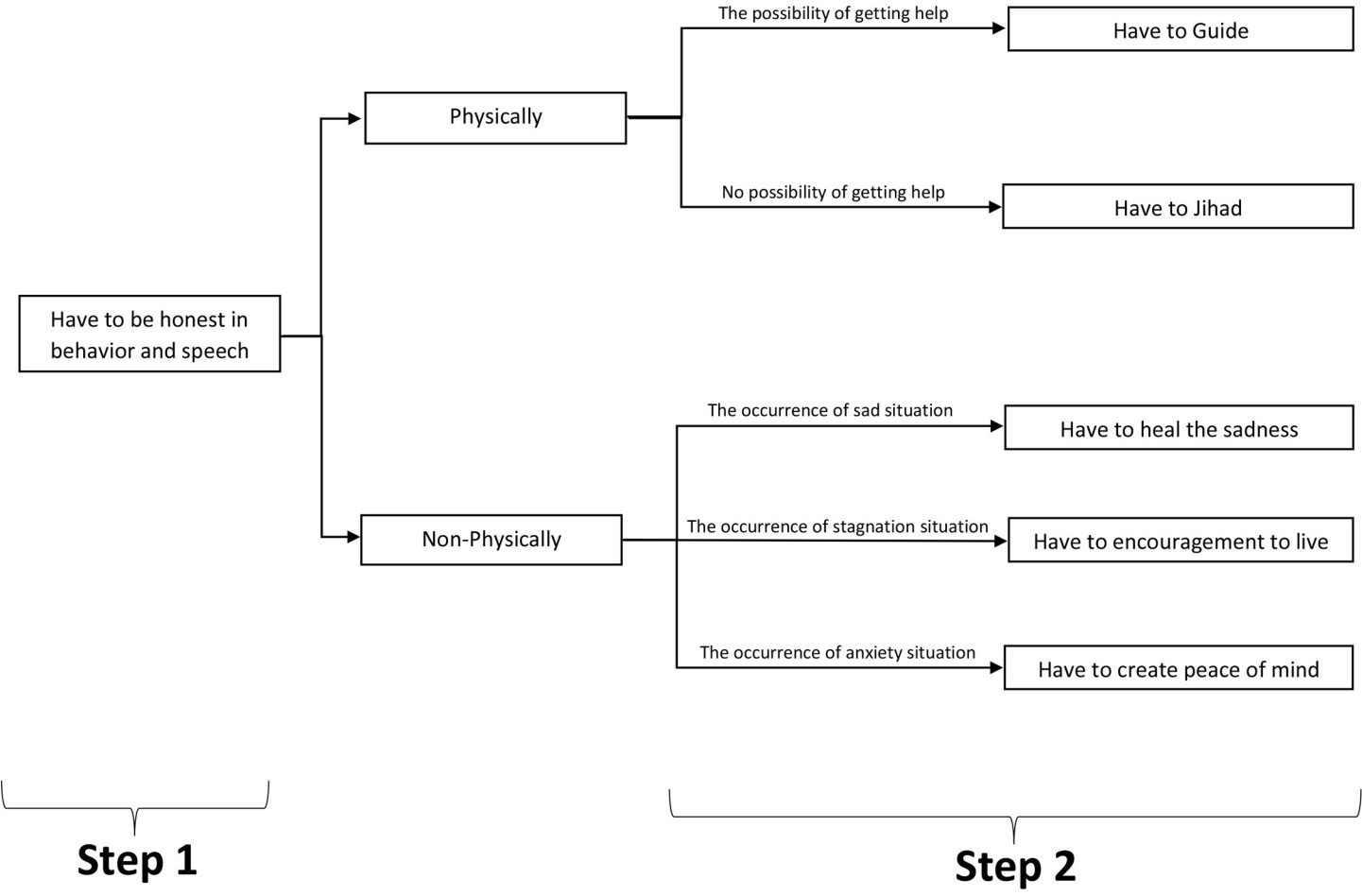

**Fig 3. Diagram of how to go through the steps.**

The laws that were approved for each procedure based on science and knowledge were accepted by nurses as the basis for their performance, and this caused them to emphasize performing their assigned tasks timely and in accordance with the law and policy. *"I know that the effect of a certain drug occurs after a certain period of time, but the doctor's prescription was something else. For example, he said to take this test at this hour and send it. I said to myself, there is no use if I take the test now; I will leave it for the same time as prescribed by scientific principles" (P13)*. The faith that existed in the nurses obliged them to be honest in their speech and behavior, although the environmental conditions could either help or hinder them in being so. Nevertheless, in the first step, nurses should have fulfilled their commitments according to their level of faith and the environmental conditions, which was to be honest in all their behavior and speech.

**To guide.** Considering that humans have two physical and non-physical aspects, in the second step, nurses had to determine their duties with regard to the patient's physical and non-physical condition (Fig 3). Nurses faced two situations in terms of the physical aspect of care. First, they could get help in providing care from the patient and the environmental conditions and facilities. In such cases, the nurses committed to choose the guidance sub-path and provide the context for the guidance of the patient. In fact, in situations where the patients and environmental conditions were helpful, the commitment of the nurses, which was to perform

the correct action for the patient, was to guide and show the way to the patients with grace and kindness. By respecting the authority of the patients, by educating the patients, nurses made it possible to influence their attitude, and then they made it possible for the patients to make a free and informed choice about their treatment process. One participant said: *"I somehow helped him to make a decision. The patient himself was also unaware of whether to do a CT scan or not. For example, he said that he had been hospitalized for three days and would do another CT scan to see if his lungs were better. If they weren't better, then there was no point in staying here and he would go home. I told him this was not a convincing reason to do a CT scan again. CT scans have side effects; it has radiation, and we use lead. I told him the advantages and disadvantages. Finally he decided to make a decision by the next day" (P5).*

In some cases, nurses performed the task of guiding by persuasive methods such as showing objective documentation and understanding about the logical cause of care. In this regard, a participant said: *"Some were anxious and dependent on oxygen, and there were also patients who did not cooperate to be given oxygen. For example, a patient was constantly walking in the corridor with severe shortness of breath. Sometimes when the workload was low, we had to explain the importance of oxygen to him and tell him he should put on his oxygen mask. Or we showed him his blood oxygen chart and said that it was too low and it should reach a normal level so that we can discharge him (P4).*

**To jihad.** Nurses also faced situations in which it was not possible to get help from the patient or environmental facilities for physical care. Naturally, no one likes to continue working in such conditions and endure more hardship, but the nurses in the present study, under the same conditions, committed themselves to choosing the sub-path of jihad, i.e., to struggle with their self-indulgence and desires (Fig 3). For example, nurses willingly endured the hardships related to wearing special plastic clothing, frequent disinfections, the management of complex patients who had to take care of several services at the same time, or doing things beyond their job description, such as staying in the hospital for more than the legal hours. In a way, they fought against their complacency. One participant said: *"Finally, the personnel were under various hardships, including clothes and families, and excruciating disinfections" (P16).*

Some patients even seriously hindered the provision of care that the nurses considered correct. The aggression of patients and their companions happened frequently due to the lack of equipment or hypoxia of the patients. The probability of anger in nurses in such a situation was very high, but the nurses fought against their selfish desire to express anger and showed patience. About jihad with her selfish desire to show anger, one participant said: *"I had seen from experience that finding a drop in saturation and the hypoxia they suffer negatively affects people's morals. That is, sometimes they cursed us. We can also curse them, but we remain silent and stay by their side" (P7).*

In some cases, the nurses noticed that the patients were concerned about paying the expenses. They knew the right thing to do in resolving this concern. They formed financial aid groups from among themselves or talked to the hospital's support department to help deserving patients. With this work, which was unique and has not been described in any article so far, the nurses fought against their selfish desire to keep property and save money. For example, the 16[th] participant said: *"I was responsible for collecting financial aid. We also had a few nurses in the ward who accompanied us, and we paid the expenses of the patients who we determined did not have financial resources" (P16).*

Some nurses witnessed the violation of the legal rights of some patients. They saw that the right thing to do was to prevent the occurrence of oppression so that justice could be implemented towards the patients. They decided not to discriminate between patients as much as they could and to protect the rights of the unconscious patient the same as the conscious patient. In relation to not oppressing the patient, one participant said: *"There were many times*

*that the laboratory lost the sample or the ESR tubes fell out of their hands and broke, and they call to send the sample again. If we send the same test again, the patient will not be charged extra" (P13).*

The culmination of the nurses' struggle with selfishness was sacrificing their lives to save the patient's life. Nurses sacrificed their most desirable carnal desire, i.e. their own lives, to be able to save a patient's life. One participant said: *"When the patient's lung was involved and he couldn't breathe, it was a bad situation, meaning you felt like he was suffocating. Anyway, you should have run with your heart and worked for him. At that time, the use of personal protective equipment was not important to us" (P8).*

Other nurses exhibited courage in determining and implementing the correct action; they were even willing to accept legal consequences and reprimands. For example, when the nurses knew that some doctors' orders were harmful to the patients, they changed them, or when the doctor's order was not effective for the patient, they tried to perform a procedure without the doctor's order. Naturally, such actions are associated with fear, but the nurses fought their fear to save the patient's life and fulfill their commitment; in this way, they showed courage. In this regard, one participant said: *"In adjusting the setting of the ventilator, the patient's doctor did not have much experience and set the ventilator in a way that was not in the patient's favor. Well, we used to change the setting. For example, according to the patient's condition, we used to say that certain FIO2 is good for the patient. Or this PEEP is good for the patient" (P2).*

In all these events, the struggle of nurses with their sensual desires is evident. The concept of jihad also means fighting against the obstacles of the path to be taken. Rarely does anyone want to take on this fight, because a person is naturally reluctant to endure hardship and suffering. Of course, there are different types of jihad, but the jihad against oneself, which includes struggle with loved ones and the desires of the self, is the best type of jihad. The faith that existed in the nurses of the present study committed them to such a struggle.

**Healing grief.** Nurses were faced with three situations in the non-physical aspect of care (Fig 3). First, some patients felt extremely lonely, homesick, and sad due to the environmental conditions such as quarantines and the prohibition of visits during the Covid-19 pandemic. In one sense, their souls were under pressure, because they liked to share their sorrows and worries with someone. Therefore, nurses committed themselves to choosing the sub-path of healing grief and comforting patients. Nurses' empathy and sympathy with patients was shown by their listening to patients and sitting by their sides. For example, one participant said: *"Sometimes the patient is alone, and she really wants to talk about her illness. As long as my time allows, I stand over her for four or five minutes and listen to her" (P13).*

The gentleness of the nurses and the use of beautiful and interesting expressions along with the emotional harmony they create with their patients also helped relieve homesickness and create happiness. Some nurses related the jokes they made with the patients that made them laugh and cheer them up. The 14th participant said: *"Many times by saying my father, my mother - you are being too naughty - take a break. We often used these words, which may be slang" (P14).*

In the current study, the nurses realized that the nature of Covid-19 caused patients' social relations to decrease drastically, and the patients were upset because of this. Nurses tried to meet and accompany the patient under different pretexts and in different situations, such as verbally, touching, or writing, so as not to harm the patient's recovery process. The fifth participant said: *"Many of the patients who were connected to ventilators used to talk to us with paper. Many of our communication techniques were written on paper. I kept many of the writings of the patients connected to ventilators. Then, if the patient died, I would give these to their companions and say that they had these wishes" (P5).*

Furthermore, the limited comfort facilities and lack of physical comfort was not ineffective in causing patients' sadness. The nurses saw the right thing to do as much as they could and the environmental facilities allowed with all kinds of methods, such as giving comfortable clothes, changing beds that are problematic for patients, allowing them to bring personal comfort items, providing comfortable means for the patients' companions to sleep or eat, or even helping patients perform religious duties. Nurses sought to create comfort and physical well-being for the patients and to help patients' companions. In this regard, the 14th participant said: *"It was very difficult for my patient to go to make ablution, so we helped him with a wheelchair or an assistant to go to the bathroom to make ablution" (P14)*. Also, about helping the companions, the 15th participant said: *"Sometimes we would give extra food to the companions" (P15)*.

**Encourage to continue living.**   Environmental conditions such as the spread and high lethality of Covid-19, the lack of a definite treatment for the disease, and the unpredictability of mortality and drug effectiveness caused some patients to suffer depression and lack motivation to continue treatment. Therefore, the second situation faced by nurses in the non-physical aspect of care was stagnation of the soul in patients and possibly deterioration of the soul's vitality. In this situation, nurses felt committed to choosing the sub-path of encouraging the patient to continue living; by promoting vitality and energy, they encouraged patients to continue living. About creating vitality in patients, one participant said: : "*For example, the patient who is very sensitive and when he sees his family, his conditions improve a lot, I would let his family wear a gown and a mask to come in and see the patient and leave*" *(P6)*.

Considering that the Covid-19 pandemic caused many deaths even among young people, injecting hope in patients to encourage them to continue the treatment process was another action that nurses considered correct in the process of fulfilling the commitment of encouraging them to continue their lives. Regarding the injection of hope, the ninth participant said: "*I used to give him a drug. For example, if there were 5 doses, I would say it's good, today is the third day, there is nothing left. I mean, I gave them hope. For example, this course is 5 days long, if you endure it for two more days, you will be fine and go home*" *(P9)*.

The existence of complexity and the strange impact of both aspects of human existence were additional distinguishing features of the Covid-19 pandemic. Each dimension of health was related to and affected each physical and non-physical aspect of care. It was decisive in the recovery process of patients. The interviews revealed that nurses used the spirituality of a patient to improve the patient's strength and ability to continue treatment. Sometimes the nurses themselves provided spiritual help to the patients, and sometimes they got help from clergymen. For example, one participant said: "*There was also the spiritual aspect that we paid attention to. Now, we were not able to do much, but the jihadist groups that came from the mosques, for example, the prayers of Imam Javad and similar things they brought, I saw that it was very effective*" *(P9)*.

As mentioned, one of the unique features of the Covid-19 pandemic was the great impact of the loss of spiritual vitality on the health of the body. The loss of spirit and losing oneself in sensitive and homesick patients caused a disruption in the healing process. Therefore, the faith that existed in the nurses committed them to doing their best in encouraging patients to live.

**Creating peace of mind.**   The third and final situation that the nurses faced in the non-physical aspect of care was that the pressure of the soul in some patients was related to anxiety and worry about the existing conditions. For this reason, nurses felt committed to choosing the sub-path of creating peace and security of mind for patients (Fig 3). In some cases, not respecting the patient's red lines, such as privacy, or not paying attention to the opinions and beliefs of patients due to limitations in environmental conditions, caused anxiety in patients. For this reason, the nurses saw the right action as being more flexible in paying attention to the patients' red lines as far as the environmental facilities allowed. For example, about

respecting the patient's privacy, the 10th participant said: *"If the patient's companions were present, we would say to stand in such a way as to create a cover or, for example, take a bed sheet or a cloth and open it to create a cover so that no one could see. We were very sensitive about this matter and paid attention to it, but we did not have good facilities" (P10)*. In relation to respecting the opinions and beliefs of the patient, the first participant said: *"The patient said that I don't want a lady to come and take blood from me. If we had a male nurse, we would tell him to come" (P1)*.

Gaining the trust of the patients and solving their mental concerns in this area were other actions that the nurses considered correct and necessary for the patient. Covid-19 differed from other diseases in that patients lacked trust in hospital treatment and care, Therefore, nurses tried to gain the trust of patients so that there would be no mental worry for patients in the continuation of the care process. They used various methods, such as showing themselves as capable in nursing, explaining the logical cause of the actions, performing the assigned tasks accurately, responding and meeting the patient's needs in a timely manner, etc., so as to play a positive role in facilitating the healing process by gaining the trust of the patients. The seventh participant said: *"Because we inject corticosteroids and they raise the blood sugar level, we also inject dexamethasone. Dexamethasone also raises your blood sugar too much. I started to tell him about the medicines. You see, when we talked to them like this, the patient would subconsciously trust us" (P7)*.

Sometimes patient anxiety was related to the unknown nature of the Covid-19 disease and the general fear and panic induced by the media and the environment. Nurses tried to resolve this anxiety by talking, diverting the patient's thoughts, more defiance, and sedative drugs. Their stress sometimes interfered with the implementation of therapeutic interventions, and this was another difference between Covid-19 and other diseases. For example, the sixth participant said: *"A lot, that is, 99% of our patients were stressed. They calmed down a lot by talking to us. After that, just being calm made their situation better" (P6)*. It can be acknowledged that nurses, moving along the turbulent path of patients with Covid-19, always tried to implement the correct action and considered it their commitment at every moment. The issue that was important was fulfilling one's commitment.

## Discussion

The current findings show that nurses, having unique personality traits, provided ethical care to patients with Covid-19. The values, experiences, and desires that exist in each person determine the degree of faith a person has, and faith is the main driver of the manifestation of the sense of obligation and commitment. In line with these results, Muaygil in 2023 reported a sense of obligation and inner commitment existed in the treatment staff for performing care for patients with Covid-19, a commitment that made the treatment staff, in the most difficult and uncertain conditions, recognize their duty and seek to do it [20]. Although both the present study and Muaygil's research revealed the existence of an inner requirement and commitment, the strength and distinction of the present study lies in the presence of the divine aspect of this inner commitment, which causes all eyes to turn to the same source, which is God's satisfaction. Values, experiences, and desires can create different duties in different people and cause various behavioral and cognitive consequences [21]. Having a divine aspect and religious beliefs makes nursing actions purposeful and meaningful and makes it easy and even enjoyable for nurses to endure problems and hardships [22]. In this regard, McKenna's 2023 study identified the main reason for nurses helping patients, even during the Covid-19 pandemic, as the existence of an inner voice, the satisfaction of God and conscience, as well as the

encouragement of those around them. Nonetheless, McKenna recommends that more research be conducted to understand how these factors affect nurses' choices [23].

As stated in the findings section, in addition to the degree of faith affecting the severity of nurses' sense of commitment, different environmental conditions also affect how and how much nurses do what they believe is right. The existence of inhibiting environmental conditions in the current study, such as the lack of equipment and nurses, along with cumbersome and sometimes inconsistent rules, which have also been stated in various studies, caused dissatisfaction and ethical tension among nurses [24, 25]. In this regard, McGuire described the challenges and ethical conflicts in the American society during the Covid-19 pandemic. The analytical results of the data indicated that most of the identified ethical conflicts were attributed to inhibiting environmental conditions, such as incompatible laws or the absence of laws [24].

Another inhibiting environmental condition in the present study was the presence of all kinds of mental, emotional, and social suffering for the nurses, which negatively impacted how they could do the right thing and their inner commitment. In line with these results, Dasilva reported in his study that the existence of social suffering and bullying by high-ranking officials and colleagues towards low-ranking nurses during the Covid-19 pandemic reduced the ethical behavior of medical staff towards patients [26].

However, the existence of some environmental conditions made help available to nurses in fulfilling their commitment and doing what they thought was right. Reducing suffering with fair management and better interactions between managers and nurses and cooperating with colleagues in performing care duties better and more easily both played a supporting role in the efforts of nurses to do what they thought was right. In line with these results, various studies have reported the type of communication between nurses and other medical personnel to be effective in improving care behavior [27, 28]; Favorable communication between the members of the treatment team can not only improve the work environment, but also reduce disputes and dissatisfaction [29] and improve the care services provided to patients [30].

The jihadist management also helped by compensating for the lack of nurses and equipment from other departments and hospitals. Jihadist management is one of the most important native styles of management in Iran, which took shape and consistency during the Islamic Revolution. Jihadist management can be considered as a work or an effort that is based on divine intention and knowledge and wisdom [31]. An example of jihadist management in the present study, which is one of the strengths and distinctions of the present study, was the help of volunteer and jihadist groups in compensating for the lack of nurses. In fact, it is a precedent that in every unfortunate event and natural disaster that occurs in Iran, volunteer and jihadist groups spontaneously act to help without waiting for orders from the government. During the Covid-19 pandemic, jihadist groups went to hospitals and helped nurses and patients. These people knowingly and selflessly risked their lives to restore health and peace to the society; Hospital managers also organized these groups [32, 33].

Hourly admission of patients to keep beds empty was another example of jihadist management and a strength of the present study, which was implemented to compensate for the lack of equipment. This action, which is unique in the world, allowed patients needing only medicine and no other service to visit the hospital on an hourly basis, receive the necessary medicine, and then go home, leaving more beds free for patients who required more care.

Analysis of the results revealed that during the process of behavior based on faith, when encountering a patient with Covid-19, in the first step, nurses considered themselves committed to doing all their actions honestly, because one of the most important dimensions of behavior based on faith is having an honest and unvarnished relationship with the audience [34]. In their study, Karimi also mentioned that honesty in behavior and adherence to professional commitments were higher among nurses who had heart-felt satisfaction and some kind of

inner commitment toward the duties assigned to them [35]. Jesmi interviewed 17 nurses about being honest with patients, and the results showed that various factors such as religious beliefs, inner voice, philanthropic feeling, empathic care, and belief in obtaining God's satisfaction and rewards made nurses act honestly in their behavior [36].

In the second step, nurses should pay attention to both physical and non-physical aspects of the patient so as to perform the necessary action and fulfill their commitment based on the condition and situation of the patient. For the physical aspect of care, nurses have two paths to choose, i.e., guidance and jihad. By choosing the path of guidance, nurses provide the context for changing patients' attitudes through knowledge enhancement. In their study, Russ considered methods such as getting help from the patient's family and important people, reminding patients of the feeling of regret that they will experience later, all members of the treatment staff advising patients at different times, consulting with a psychiatrist, and expressing the benefits of the procedure as effective in increasing the knowledge of patients and changing their attitude [37].

The second path that exists in the physical order is jihad, or struggle, with the existing obstacles and conditions to provide the right and necessary care. In the present study, sometimes the jihad was with the desires of the self, such as the desire to express anger, seek comfort, or avoid suffering, and sometimes the jihad was with such desires as loving life and money more than saving the patient's life. In line with these results, in their 2023 study, Slettmyr stated the sacrifices of nurses in enduring turbulent conditions and adapting themselves to any situation despite anxiety and fatigue as well as facing challenges and ethical conflicts in the Covid-19 pandemic [38]. Despite the results of Slettmyr's study, the nurses participating in Ciezar-Andersen's study reported that sacrifice in the nursing profession is the result of the existence of a dominant stereotype image of the "ideal nurse," which leads to job dissatisfaction, constant availability, and job burnout in nurses, so sacrifice is an irrational thing [39]. The presence of conflicting studies in this field is proof of the claim that the degree of faith is effective in identifying a nurse's duty and commitment. Sacrifice is seen as an ethical act and inner commitment in some nurses, and in others, it is a stereotypical image that leads to job burnout.

In addition to the sacrifice of nurses, one of the surprises of the present study and the unique initiatives of nurses was giving up their own money to help patients, which has not been reported in any study so far. In the present study, financial aid meant donating money for God's satisfaction, generally jihad against selfish desires, and this act has a direct relationship with the beliefs and values of nurses and is generally related to their degree of faith and sense of commitment.

In addition to the physical aspect, the Covid-19 disease also affected the non-physical aspects of patients; therefore, while providing optimum physical care for patients, the nurses also tried to relieve the pressure on the souls of the patients. In the non-physical aspect of care, there are three paths in front of nurses. For patients who were in a situation of grief, nurses considered themselves committed to using any method, including gentleness while talking with the patient or emotional compatibility, to heal the patients' grief. In this regard, Stylianou stated that in some situations, the patient's grief may be very complicated, such that the nurse does not know what to say or how to talk to someone who is sad. Nevertheless, the most important issue is that the nurse is present, listens to the patient's words, and allows the patient to share their feelings with the nurse [40]. Another method of healing grief was to create physical well-being for the companions of the patients in addition to the patients themselves. Patient companions are able to promote the patients' happiness and satisfaction with the quality of the care received through the transmission of their positive feelings about the facilities and welfare services received [41].

The second way that exists in the non-physical aspect is to encourage patients to continue living. If the patients were in a state of soul stagnation, and there was a possibility of them digressing and becoming depressed, the nurses felt committed to using various methods such as energizing, injecting hope, and seeking help from the patient's spirituality. In line with these results, Ribeiro stated in their 2022 study that having hope and being optimistic can play a major role in the patient's recovery process [42]. The results of Yıldırım's study also showed a significant positive relationship between giving hope and vitality of the soul with the well-being of Covid-19 patients. Yıldırım recommended that special attention be paid to improving the soul vitality of patients using creative methods, especially in critical situations such as the Covid-19 pandemic [43].

In the present study, despite the presence of some patients in a state of sadness or soul stagnation, due to the unknown nature of the disease, some patients were also in a state of anxiety and worry. As soon as the nurses noticed the presence of anxiety in the patients, they felt committed to take action, such as talking, diverting the patient's thoughts, showing more defiance, and giving sedatives, to create peace and security for them. In this regard, Xi stated that it should not be assumed that the level of peace of mind in patients remains constant with the passage of time. In their descriptive study, they proved that if no action is taken to maintain or improve the peace of mind of patients, with the passage of time, peace of mind fades and the patient becomes confused. For this reason, Xi considered the lack of attention to the state of the souls of patients in the Covid-19 pandemic as a serious gap and recommended that more studies be done in this field [44]. In addition to proper and necessary physical care, the nurse's companionship and kindness in relieving pressure on the soul is one of the most important factors in the patient's recovery process; kindness and compassion towards the patient not only can be effective in the patients' perception of the presence of instrumental support from the nurses [45], but also create peace of mind and reduce anxiety and aggressive behaviors in patients [46].

## Limitations and recommendations

The present study was conducted among nurses in teaching hospitals, and results may differ with nurses working in private hospitals. Therefore, conducting future studies in private hospitals and exploring other diverse contexts can lead to a better understanding of the ethical care process of these patients and ultimately make the ethical care model more comprehensive and effective. The current study was also limited in its access to the participants under the conditions of the Covid-19 disease, by the lack of free time among nurses, and nurses' fatigue caused by the increase in workload during the Covid-19 pandemic. To rectify this limitation, a web-based interview in addition to face-to-face was used. The third limitation was a lack of access to nurses at managerial levels such as supervisors. Fortunately, after long-term follow-up, three people were finally interviewed. The last limitation was the presence of the first researcher in the hospital to observe the care, which was associated with difficulty in coordination, and this limitation was finally resolved with the help and guidance of the supervisor.

## Conclusion

The present study has explained the process of ethical care of patients with Covid-19 in detail. The results indicated that starting the process with a problem means a challenge of how to do the right or correct thing for the patient. Nurses are mainly drive to manage the problems by their feeling of commitment to do the right and necessary action for the patient. Commitment is an inner matter and not an external requirement. Thus, a force from within, which is faith, manifests the feeling of commitment. Because the intensity and degree of the sense of

commitment differ based on the degree of one's faith, fulfillment of commitment is graded from zero to infinity. The actions that a nurse takes to fulfill her sense of commitment is called the faith-based behavior process. The process of behavior based on faith depends on the context. In addition to being influenced by the personal characteristics of the nurse, it is influenced by environmental conditions that are sometimes helpful and sometimes hindering. The process of behavior based on faith is done through two steps. In the first step, the nurse considers herself committed to being honest in all her behaviors and speech. In the second step, the nurse considers herself committed to paying attention to both physical and non-physical aspects of the patient. For the physical aspect, the nurse should determine whether she can get help in providing care from the patient and the environmental conditions. Based on this, she considers herself committed to choosing one of the sub-paths of guiding or jihad to provide care. In the non-physical aspect, the nurse considers herself committed to choosing the sub-path of healing grief for patients experiencing this emotion. If the patient is in a state of soul stagnation, the nurse considers herself committed to choosing the sub-path of encouraging the patient to continue living. Finally, if the patient is in a state of anxiety and worry, the nurse considers herself committed to choosing the sub-path of creating peace of mind. Considering the existence of the conscientious root of commitments as well as the addition of the divine attribute to it, the ultimate goal of doing the right action and fulfilling commitments by nurses can be considered to be the satisfaction of God as well as the satisfaction of conscience. Therefore, the more a nurse fulfills her commitments, the closer she gets to the final destination, which is the satisfaction of God and conscience. The theory of "faithful nursing" can be used as a guide to describe and expand the role of nurses in providing ethical care, developing clinical guidelines, and planning educational programs for nursing students and nursing staff.

## Supporting information

**S1 Table. The first version of interview guide.**
(DOCX)

**S1 File. Primary codes.**
(DOCX)

## Acknowledgments

This paper is part of a PhD dissertation in nursing which was approved by the Medical Ethics Committee of Shahid Beheshti University of medical sciences, Tehran, Iran. The authors would like to thank Vice Chancellor for Research of Shahid Beheshti University of Medical Sciences and all those who helped us with our research.

## Author Contributions

**Conceptualization:** Hamideh Azimi.

**Project administration:** Fariba Bolourchifard.

**Validation:** Fariba Borhani.

**Writing – original draft:** Rafat Rezapour-Nasrabad, Akram Sadat Sadat Hoseini.

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
