## [Decision Letter · Decision Letter 0]

14 Sep 2023

PONE-D-23-17837Study of the process of ethical care in patients with covid-19: A grounded theoryPLOS ONE

Dear Dr. Nasrabad,

Thank you for submitting your manuscript to PLOS ONE. After careful consideration, we feel that it has merit but does not fully meet PLOS ONE’s publication criteria as it currently stands. Therefore, we invite you to submit a revised version of the manuscript that addresses the points raised during the review process.

Please submit your revised manuscript by Oct 29 2023 11:59PM. If you will need more time than this to complete your revisions, please reply to this message or contact the journal office at plosone@plos.org. Please include the following items when submitting your revised manuscript:A rebuttal letter that responds to each point raised by the academic editor and reviewer(s). You should upload this letter as a separate file labeled 'Response to Reviewers'.A marked-up copy of your manuscript that highlights changes made to the original version. You should upload this as a separate file labeled 'Revised Manuscript with Track Changes'.An unmarked version of your revised paper without tracked changes. You should upload this as a separate file labeled 'Manuscript'.

We look forward to receiving your revised manuscript.

Kind regards,

Myriam M. Altamirano-Bustamante

Academic Editor

PLOS ONE

“We are heartily grateful to the research council of the Shahid Beheshti University of Medical Sciences for financially supporting the study.”

Additional Editor Comments:

It is a potential big paper but require a lot of work to reach the top level.

Please follow the observations of the reviewers.

Reviewers' comments:

Reviewer's Responses to Questions

**Comments to the Author**

1. Is the manuscript technically sound, and do the data support the conclusions?

Reviewer #1: Yes

Reviewer #2: Partly

2. Has the statistical analysis been performed appropriately and rigorously? 

Reviewer #1: N/A

Reviewer #2: N/A

3. Have the authors made all data underlying the findings in their manuscript fully available?

Reviewer #1: Yes

Reviewer #2: Yes

4. Is the manuscript presented in an intelligible fashion and written in standard English?

Reviewer #1: Yes

Reviewer #2: No

5. Review Comments to the Author

Reviewer #1: PONE-D-23-17837 Study of the process of ethical care in patients with covid-19: A grounded theory

1. The study presents the results of original research.

Yes, this is original research. The introduction leads me as a reader forward to the aim.

2. Results reported have not been published elsewhere.

Yes

3. Experiments, statistics, and other analyses are performed to a high technical standard and are described in sufficient detail.

High technical standard and in sufficient detail, but there is a need of clarification about the recruitment procedure, by whom? Please clarify theoretical sampling and saturation all readers are not familiar with these terms. Interview guide revised after every interview? It should be after every analysis—that is when you know if there is something new to focus on. Information about the interview guide? and pilot interviewing was it performed? Please, clarify and give us some more information.

How come that there were 21 interviews, but 19 participants?

After 19 interviews there was no new data, meaning that the categories were filled and stable?? No new codes or categories were identified.

There was a single researcher performing the interviews, how did this person manage to make observations and taking notes at the same time?

If taking/spending shift as a nurse it is hard to do observations. Were you there as an observer?

The analysis is presented according to the book. However, there was no axial coding?

Trustworthiness is presented, according to reference 27. Grounded theory do not us these concepts (since it does not present itself as an ordinary qualitative method)- since it has its own quality criteria. Corbin and Strauss present 8 criteria.

Strauss and Corbin do clarify how credibility, trust, and applicability etc are to be managed in GT.

The result- information about the participants can be placed in the methodology section—sample (free of choice). Decide if the information should be in text or in table, now it is duplicated. Still, confusion about 21 interviews and only 19 participants.

The result presentation would benefit from be presented below table 1.

In total 10…… bridging over to the presentation: Main concern

The result presentation is a bit confusing—the result in a Straussarian GT is an integrated theoretical structure. The main category in this structure is??? Main concern about what??

Then there are four categories presenting strategies- clarified by the sub-categories….

Table 2 make me even more confused—Here the main categories are -Context, Strategies and outcome??? I am not sure what this table 2 is used for. In GT there you have a matrix as a tool in the coding process- otherwise a figure to demonstrate the result.

The study aimed to explain the ethical care process of patients with COVID-19. Please, present a result according to Straussarian GT answering your aim.

The main Category: Striving to create the most ethically valuable care

A suggestion is to use the Figure 1 as the starting point and present the findings and explanation from that.

Why are there quotations piled when presenting the categories? This is not the way to present a GT study.

Please keep focus, the issue under study is about nurses’ ethical concerns in Covid 19 care.

There is not much of these perspectives in the result…

The result section needs to be re-written. This study is focusing on an important and interesting area of research, so please clarify and structure the result. Let the figure guide the writing not necessary state every part- especially when there is only one sentence.

The discussion starts with another aim, providing ethical care to patients with Covid-19 in order to achieve the desired level of health.

Quite a lot of the text in the discussion is information I would have found in the result. Please move it and clarify the result, then it will be clear to understand the discussion and all the 49 references added in the discussion.(are they all needed?)

There is also an error, in a study conducted by Pazokian in 1401, in a study conducted by Basouli in 1400???

Limitations are well presented but are lacking methodology issues. However, there are several GT studies conducted about ethical care/approaching. Sorry, but this is not the first one. Qualitative studies are not aiming for comparison, but adding and deepening knowledge.

4. Conclusions are presented in an appropriate fashion and are supported by the data.

Conclusions are presented in appropriate fashion, probably supported by data, but as mentioned previously, the result need to be clarified. I do not agree that the process of providing ethical care to patients with Covid-19 was explained in detail. But this could be corrected.

5. The article is presented in an intelligible fashion and is written in standard English.

Yes, the article is presented in an intelligible fashion, but the structure could be sharpened, and it is written in standard English, mostly. (the text could be more straight forward)

6. The research meets all applicable standards for the ethics of experimentation and research integrity.

Yes, this study meets all the applicable standards for research integrity.

7. The article adheres to appropriate reporting guidelines and community standards for data availability.

Yes, the article is following the reporting guidelines.

This could be an interesting paper, presenting important knowledge. This paper needs to be clarified, re-written result section and a straighter forward discussion, focusing on the results and how it answers the aim.

Out of 77 references 18 are 10 years old or more (4 are 9 years old), there are 26 references in the introduction and out of these 7 are 10 years old or more.

Reviewer #2: Thank you for your scholarship in this area. I would offer that you have not yet reached the threshold of a grounded theory. The core category "create the most ethically valuable care" does not illustrate the process of how healthcare professionals created ethical care in their work. The core category should tell us what people did to make the process happen.

Additionally, the sub categories of jihad, guide, respect, and have compassion and love are descriptive, and do not indicate a process. In grounded theory, the order of the themes matters, showing that a participant moves from one to the next. There are also too many sub themes, indicating there has not yet been enough synthesis and the data needs additional analysis.

I suggest that you could do some additional analysis and refine the theory, then resubmit that that time. You could look at examples from other grounded theories:

Jackson, J., Vandall-Walker, V., Vanderspank-Wright, B., Wishart, P., & Moore, S. L. (2018). Burnout and resilience in critical care nurses: A grounded theory of Managing Exposure. Intensive Crit Care Nurs, 48, 28-35. https://doi.org/10.1016/j.iccn.2018.07.002

Reyes, A.T., Andrusyszyn, M.A., Iwasiw, C., Forchuk, C. & Babenko-Mould, Y. (2015) Nursing students' understanding and enactment of resilience: a grounded theory study. Journal of Advanced Nursing 71(11), 2622–2633. doi: 10.1111/jan.12730

I also suggest an edit from a native English speaker prior to resubmission.

6. PLOS authors have the option to publish the peer review history of their article (what does this mean?). If published, this will include your full peer review and any attached files.

Reviewer #1: No

Reviewer #2: No

---

## [Author Response · Author response to Decision Letter 0]

1 Nov 2023

Hello Dr. Myriam

The reviewers' excitement for the article and the insightful criticism they and the Chief Editor provided are both sincerely appreciated. The manuscript's readability and validity have been greatly improved by these observations.

I edited the article based on the reviewer's comments. Also I responded to all comments.

Please see the table in response to reviewers file.

Gratitude once more.

---

## [Decision Letter · Decision Letter 1]

19 Feb 2024

Ethical care in patients with Covid-19: a grounded theory

PONE-D-23-17837R1

Dear Rafat Rezapour-Nasrabad

We’re pleased to inform you that your manuscript has been judged scientifically suitable for publication and will be formally accepted for publication once it meets all outstanding technical requirements.

Kind regards,

Myriam M. Altamirano-Bustamante

Academic Editor

PLOS ONE

Additional Editor Comments (optional):

Reviewers' comments:

Reviewer's Responses to Questions

**Comments to the Author**

1. If the authors have adequately addressed your comments raised in a previous round of review and you feel that this manuscript is now acceptable for publication, you may indicate that here to bypass the “Comments to the Author” section, enter your conflict of interest statement in the “Confidential to Editor” section, and submit your "Accept" recommendation.

Reviewer #1: All comments have been addressed

2. Is the manuscript technically sound, and do the data support the conclusions?

Reviewer #1: Yes

3. Has the statistical analysis been performed appropriately and rigorously? 

Reviewer #1: N/A

4. Have the authors made all data underlying the findings in their manuscript fully available?

Reviewer #1: No

5. Is the manuscript presented in an intelligible fashion and written in standard English?

Reviewer #1: Yes

6. Review Comments to the Author

Reviewer #1: Thank you for all efforts in amending this manuscript. All comments and suggestions from the reviewers have been taken into consideration. I am looking forward to reading this article when it is published.

7. PLOS authors have the option to publish the peer review history of their article (what does this mean?). If published, this will include your full peer review and any attached files.

Reviewer #1: No

---

## [Editor Report · Acceptance letter]

20 Mar 2024

PONE-D-23-17837R1 

PLOS ONE

Dear Dr. Rezapour-Nasrabad, 

I'm pleased to inform you that your manuscript has been deemed suitable for publication in PLOS ONE. Congratulations! Your manuscript is now being handed over to our production team.

Kind regards, 

on behalf of

Dr. Myriam M. Altamirano-Bustamante 

Academic Editor

PLOS ONE